# Incidence and risk factors of perioperative respiratory adverse events in pediatric surgical patients: Development and validation of a predictive model in Brazil

Isabela S. Sirtoli [1,2], Stela J. Castro[3], Paulo C. S. Neto [1], Cleiton Pando[1,2], Clarissa Mendanha[1,2], Raissa M. Scorsatto[3], Luciana C. Stefani [2,4]*

1 School of Medicine, Universidade Federal do Rio Grande do Sul (UFRGS), Porto Alegre, Brazil, 2 Anaesthesia and Perioperative Medicine Service, Hospital de Clínicas de Porto Alegre (HCPA), Porto Alegre, Brazil, 3 Institute of Mathematics and Statistics, Universidade Federal do Rio Grande do Sul (UFRGS), Porto Alegre, Brazil, 4 Department of Surgery, School of Medicine, Universidade Federal do Rio Grande do Sul (UFRGS), Porto Alegre, Brazil

* lpstefani@hcpa.edu.br

## Abstract

### Background

Enhancing the safety of paediatric patients undergoing surgery is crucial, particularly in settings with limited resources and fragmented health systems. This study focused on investigating perioperative respiratory adverse events in Brazil.

### Objectives

To determine the incidence and associated risk factors of perioperative respiratory adverse events in pediatric surgical patients, and develop a predictive model to improve bedside decision.

### Design

Observational cohort study.

### Setting

Two high complexity public teaching hospitals in Southern Brazil.

### Patients

Patients under 16 years undergoing elective or nonelective procedures between August 2020 and February 2022.

### Main outcome measures

The primary outcome was perioperative respiratory adverse events. Patients were prospectively followed during surgery and recovery stay. Preoperative data were

**Data availability statement:** The data underlying the results presented in the study are available from Zenodo (https://zenodo.org/records/18600762).

**Funding:** This work was supported by Financiamento de Incentivo à Pesquisa do Hospital de Clínicas de Porto Alegre [FIPE-HCPA Project 2020-0363] and and the Foundation for Research of the State of Rio Grande do Sul – FAPERGS - PPSUS 2020 [Secretaria de Saúde do Estado do Rio Grande do Sul – SES-RS, Ministério da Saúde – MS – Departamento de Ciência e Tecnologia da Secretaria de Ciência, Tecnologia, Inovação e Insumos Estratégicos em Saúde – Decit/SCTIE/MS and Conselho Nacional de Desenvolvimento Científico e Tecnológico – CNPq]. URL: https://fapergs.rs.gov.br/, https://www.gov.br/cnpq/pt-br. There was no additional external funding received for this study." in your updated Funding Statement. The funders had no role in study design, data collection and analysis, decision to publish, or preparation of the manuscript.

**Competing interests:** The authors have declared that no competing interests exist.

collected through interviews with parents and legal guardians, and multilevel logistic regression models were used for analysis.

## Results

Among 1339 children, 214 (15.9%) experienced perioperative respiratory adverse events. Desaturation was the most frequent respiratory complication, occurring in 111 children (8.3%), followed by laryngospasm (84, 6.3%). The final risk model exhibited good discrimination with an AUROC of 0.71 (95% CI 0.68-0.75) and had superior accuracy (AUROC, 0.69 vs. 0.62; p = 0.004) compared to the COLDS score in patients under 6 years. Variables included in the model were age < 1 year, current upper respiratory infection, history of prematurity, lung or airway disease, and the interaction between tracheal intubation and airway surgery.

## Conclusions

The study identifies a high incidence of perioperative respiratory adverse events (PRAE) in Brazilian pediatric surgical patients and key risk factors. A novel risk prediction model for children under 16 outperforms existing tools. These findings are vital for resource-limited settings. Future research should validate these results in varied healthcare contexts, develop targeted interventions, and assess the model clinical effectiveness to enhance pediatric surgical care and patient safety.
**Study registration:** UTN code U1111-1274–8584

## Introduction

The need for safe pediatric anesthesia care in constrained resources scenarios is urgent. Recent data show that approximately 1.7 billion children and adolescents worldwide lack access to surgical care [1] with a significant proportion residing in low to upper middle-income countries, where children make up about 50% of the population [2].

The variability of preoperative goals and perioperative decisions leads to fragmented perioperative support, backlog surgeries, and strains on already overstretched health systems [3]. Improved identification of high risk patients could enhance care and ground pathways to prevent adverse events arising from the complex interplay between systemic factors (such as access, structure, processes), surgical particularities and patient-specific factors [4]. Ultimately, risk stratification can ensure the delivery of safe, timely, and affordable surgeries [5,6].

Pediatric anesthesia has many peculiarities, including time constraints, and uncertainties that hamper decision-making and increase stress among surgical teams. The majority of complications involve the respiratory system, manifesting as stridor, desaturation, laryngospasm, bronchospasm, and hypoventilation [7–9]. Perioperative respiratory adverse events (PRAEs) are the leading cause of morbidity and mortality during pediatric anesthesia [3,10–14]. Furthermore, these complications disrupt perioperative processes, prolong hospital stays, and increase the utilization of resources [15–17].

While studies in high-income countries have identified factors associated with increased PRAEs, such as age, pulmonary diseases, upper respiratory infection (URI), intubation, and history of prematurity [3,11,18,19] there remains a lack of data on these outcomes in resource-constrained settings.

We conducted a prospective observational multicenter cohort study aiming to determine the incidence and identify risk factors associated with perioperative respiratory adverse events among children under 16 years undergoing noncardiac surgeries at two large public hospitals in southern Brazil. Additionally, our goal was to construct and validate a risk model for perioperative respiratory adverse events in a middle-income population to provide insights that enhance perioperative pathways and inform daily decision-making processes for clinicians in countries like ours.

## Methods

### Data source and study population

Our study involved a prospective cohort of pediatric patients undergoing surgical procedures from August 24th 2020 to February 22th 2022 in two hospitals in southern Brazil – from August 25th 2020 until February 22th 2020 at Hospital de Clínicas de Porto Alegre and from October 29th 2020 until February 22th 2022 at Hospital da Criança Conceição/Grupo Hospitalar Conceição. Both hospitals treat high-complexity cases within the public health system. Hospital da Criança Conceição is a pediatric hospital with 203 ward beds and 71 intensive care unit (ICU) beds, and Hospital de Clínicas de Porto Alegre is a general hospital with 88 pediatric ward beds and 33 pediatric ICU beds.

Both research ethics committees approved the study and registered it in the Brazilian Registry of Clinical Trials/REBEC (UTN code: U1111-1274–8584). We adhered to the STROBE [20], TRIPOD [21] and TRIPOD-AI [22] guidelines for reporting observational studies developing and validating multivariable prediction models.

The inclusion criteria for the study were patients under the age of 16 undergoing noncardiac surgeries and procedures. We excluded children who required anesthesia for organ transplants, were already intubated or tracheostomized prior to surgery, were undergoing procedures exclusively under local anesthesia or had missing outcome data. For children who underwent more than one surgical procedure, only the first procedure was included in the database. We followed the patients from the pre-anesthetic evaluation until two hours in the postanesthetic care unit (PACU) or ICU; patients who presented any PRAE were followed for 24 hours through medical record review.

### Ethics

Ethical approval for the study was obtained from two ethics committees. Parents or legal guardians provided informed written consent for participation. The ethics committee of the Hospital de Clínicas de Porto Alegre, under protocol number 34722720.6.0000.5327, at the address Ramiro Barcelos, 2360, Porto Alegre – Rio Grande do Sul, Brazil, coordinated by Têmis Maria Felix, on August 24, 2020. The ethics committee of the Grupo Hospitalar Conceição, under protocol number 34722720.6.3001.5530, at the address Francisco Trein, 326, Porto Alegre – Rio Grande do Sul, Brazil, coordinated by Daniel Demétrio Faustino da Silva, on October 28, 2020.

### Outcome definition

The primary outcome was a binary composite, the presence or absence of any perioperative respiratory adverse events, defined as oxygen desaturation <90%, stridor, bronchospasm, laryngospasm, or bronchial aspiration from anesthesia induction until two hours after surgery (S1 Table).

Unfavorable endpoints associated with perioperative respiratory adverse events included reintubation, unplanned hospital admission, unplanned ICU admission, bradycardia requiring atropine or adrenaline, intubation for more than 24 hours postoperatively, aspiration pneumonitis, pulmonary oedema and cardiac arrest.

## Explanatory variables

We collected data on anesthesia techniques and surgical procedures through interviews with parents or legal guardians, anesthesia documents, and patient electronic charts. Clinical data included age, American Society of Anesthesiologists (ASA) physical status (ASA-PS) classification, lung or airway disease, symptoms of upper airway infection (URI), passive smoking at home, and history of prematurity. We classified URI as either current or recent (occurring during the last six weeks). Intra-operative variables included the type of airway device, anesthesia induction method (inhalation or venous), type of anesthesia maintenance and use of neuromuscular blockers. We categorized procedures by their nature (elective or nonelective) and type (airway or general procedure) (S1 Table). There was no blinding in the assessment of predictors or outcomes.

## Sample size

Using SAS Studio, we calculated the sample size required to identify the likelihood of upper airway infection associated with pulmonary complications, adjusted for the following covariates and their respective prevalence in our population found in a pilot study: passive smoking (17%), URI (22%), prematurity (12%) and ASA Physical-State. The calculation considered a prevalence of perioperative respiratory adverse events of 15%, 80% statistical power, and a 5% significance level. The number of participants was estimated at 629, considering an effect measure/odds ratio (OR) equal to 2. However, we collected most of the data during the COVID-19 pandemic, and because the prevalence of URI in the pediatric population was considerably lower at that time, we increased the sample size by extending the period and thus obtained a more accurate post-pandemic sample of patients.

## Statistical analysis

We summarized the descriptive categorical data to absolute and relative frequencies and presented continuous data as the mean and confidence intervals (95%). We performed univariate analyses to identify possible associations between priori-defined predictor variables and the primary outcome.

For the model development, we primarily performed a multilevel logistic regression model with the hospital as the only explanatory variable (including the hospital as a random effect). Afterwards, we included variables known to impact PRAEs in successive models based on clinical plausibility and according to the conceptual model that describes the relationship between risk factors [23]. For all models, we used forced simultaneous entry rather than automated stepwise selection and included patient characteristics and factors related to surgery and anesthetic techniques. In parallel, since there were only two hospitals, we built the same models considering the hospital as a fixed effect, to compare the relevance of hospital effect in both strategies.

Analyses were performed using the Statistical Analysis System (SAS Studio 9.4) and R (version 3.5.1.) programs demonstrated the magnitude of each variable related to the outcome with OR and 95% confidence intervals. The significance level was 5%.

## Model estimation and performance

We took successive steps to evaluate the performance of the final model and its validity. We visually assessed the final model calibration by plotting the observed vs. predicted complications. We measured the model#39;s overall performance with the Brier score [24] and verified goodness-of-fit with the Hosmer–Lemeshow test. The area under the receiver operating characteristic (AUROC) curve was used to access the discrimination with the following cut-off points: AUROC of <0.7 to indicate poor performance, 0.7 to 0.9 moderate, and >0.9 high performance [25,26]. The final model was internally validated and calibrated using the bootstrap method for the entire cohort (1000 resampling interactions) [27]. Its performance was compared with the COLDS score, a validated tool built for the same purpose [19]. We evaluated the accuracy and differences between the ROC curves with the DeLong test [28].

## Results

During the 18 months of analysis, we evaluated 1339 patients. The mean age of the patients was 5.1 years (SD 4.1). Table 1 describes the characteristics of the surgical and anesthesia procedures and the medical histories of the two groups of children, those with and without PRAEs. Of 1339 patients, 214 (15.9%) developed a PRAE, and these children were more likely to be younger and have pulmonary diseases, history of prematurity, and current URI. Additionally, patients with PRAE underwent airway surgeries more frequently.

Desaturation was the most frequent respiratory complication, occurring in 111 patients (8.3%), followed by laryngospasm (84/214-6.3%), bronchospasm (37/214-2.8%), stridor (19/214-1.4%) and bronchial aspiration (7/214-0.5%) (Table 2). Unfavorable outcomes, such as reintubation, bradycardia, unplanned hospitalization, ICU admission or prolonged intubation, occurred in 10.2% of patients with PRAEs.

**Table 1. Patient clinical characteristics stratified by entire sample, patients with and patients without perioperative respiratory adverse events. Data are presented as number/total number (%) of patients.**

| | All children (%) N = 1.339 | Without PRAE (%) N = 1.125 | With PRAE (%) N = 214 | p-value* |
|---|---|---|---|---|
| **Hospital** | | | | 0.001 |
| HCC/GHC | 883 (65) | 721 (64) | 162 (75) | |
| HCPA | 456 (34) | 404 (35) | 52 (24) | |
| **Age** | | | | <0.001 |
| < 1 y | 185 (13) | 129 (11) | 56 (26) | |
| 1-5 y | 511(38) | 420 (37) | 91 (42) | |
| 5-10 y | 375 (28) | 328 (29) | 47 (21) | |
| 10-16 y | 268(20) | 248 (22) | 20 (9.3) | |
| **ASA-PS** | | | | <0.001 |
| 1 | 513 (38) | 455 (40) | 58 (27) | |
| 2 | 536 (40) | 447 (39.7) | 89 (41) | |
| 3 | 277 (20) | 214 (19) | 63 (29) | |
| 4 | 13 (1) | 9 (0.8) | 4 (1) | |
| **Lung or airway disease** | 303 (22) | 221 (16) | 82 (38) | 0.001 |
| **Obesity** | 52 (3) | 46 (4) | 6 (2) | 0.372 |
| **Current URI** | 61 (4) | 39 (3) | 22 (10) | <0.001 |
| **URI < 6 weeks** | 157 (11) | 123 (10) | 34 (15) | 0.039 |
| **Passive smoking** | 359 (26) | 299 (26) | 60 (28) | 0.659 |
| **History of prematurity** | 233 (17) | 162 (14) | 71 (33) | <0.001 |
| **Airway surgery** | 301 (22) | 234 (20) | 67 (31) | 0.001 |
| **Emergency surgery** | 311 (23) | 246 (21) | 65 (30) | 0.006 |
| **Premedication before surgery#** | 306 (25) | 257(25) | 49 (25) | 0.936 |
| **Inhalation induction** | 410 (30) | 338 (30) | 72 (33) | 0.294 |
| **Airway device** | | | | |
| Endotracheal tube | 866 (64) | 705 (62) | 161 (75) | <0.001 |
| Supraglottic device | 175 (13) | 159 (14) | 16 (6) | 0.008 |
| Facial mask | 298 (22) | 261 (23) | 37 (17) | 0.057 |
| **Topical laryngeal anesthesia** | 149 (11) | 119 (10) | 30 (14) | 0.090 |

*# Denominator is based on patients with valid data. All children n = 1191; children without PRAE n = 1002; children with PRAE n = 189*

HCC/GHC, Hospital da Criança Conceição/Grupo Hospitalar Conceição; HCPA, Hospital de Clínicas de Porto Alegre; ASA-PS, American Society of Anesthesiologists physical status; URI, Upper Respiratory Infection. *Student's t-test and $\chi^2$ test were used as appropriate.

**Table 2. Number of patients with perioperative respiratory adverse event (PRAE) and their percentage from the entire sample and from the PRAE group.**

| Perioperative respiratory events | Patients with PRAE. N | % of each complication in all patients | % of each complication in PRAE group |
|---|---|---|---|
| Oxygen desaturation | 111 | 8 | 51 |
| Laryngospasm | 84 | 6 | 39 |
| Bronchospasm | 37 | 2 | 17 |
| Stridor | 19 | 1 | 8 |
| Bronchial aspiration | 7 | 0.5 | 3 |
| Total | 214* | 15* | 100* |

*Note there were more than one event in the same patient.

## Baseline logistic regression model for PRAE

We iteratively constructed our final model by progressively incorporating predictors. Initially, we examined two methodologies: one leveraging the hospital as a random effect (mixed model S3 Table) and the other treating it as a fixed effect predictor (fixed model). As we introduced clinical, surgical, and anesthetic predictors into both approaches, we observed a diminishing relevance of the hospital predictor to the outcome. Subsequently, we rigorously excluded variables lacking statistical significance ($p < 0.05$) to establish a parsimonious model (Table 3).

The final model accounted for patient, anesthesia and procedure factors (Table 3). Patients with history of prematurity (OR 2,56 95% CI 1.80–3.63), those aged less than one year (OR 2,08 95% CI 1.41–3.05), those with lung or airway disease (OR 2.17 95% CI 1.55–3.05) and those with URI (OR 3,70 95% CI 2.06–6.66) were more likely to have a PRAE. We also tested the interaction between variables. There was a significant interaction between intubation and airway surgery. When undergoing general, non-airway surgeries, intubated patients had increased odds of PRAEs (OR 2,13, 95% CI 1.71–2.55). Likewise, among all non-intubated patients, those undergoing airway surgeries had a higher risk of PRAEs (OR 3.15, 95% CI 2.41–3.91).

## Model performance measures

The AUROC for PRAEs in the cohort was 0.71 (95% CI 0.68-0.75) (Fig 1). The Hosmer–Lemeshow goodness-of-fit statistic was 3.06 (p = 0.801), which reflected an acceptable model calibration. Additionally, the Brier score result of 0.12 (95% CI 0.11-0.12) confirmed its excellent overall performance. Fig 2 shows the calibration plot.

We obtained a sensitivity of 0.66% and specificity of 0.68% for the adjusted model, considering a cutoff value of 0.15 for PRAE (S2 Table). This value was the probability of PRAE that equalized sensitivity and specificity in the adjusted model [29].

## Internal validation

To examine the variability of our clinical prediction model we generate a bootstrap sample and developed a bootstrap prediction model replicating the same model building strategy as used originally. The bootstrapping procedure for internal validation in 1000 bootstrap samples provided AUROCs varying from 0.70 to 0.71, demonstrating good predictive capacity. The c-statistic corrected for optimism was 0.70, with an average optimism of 0.01. The Brier score adjusted for optimism was 0.12.

We summarized the instability in the model#39;s predictions by calculating the average Mean Absolute Predictor Error (MAPE) [30] which was 0.0005920504 (the lower the MAPE value, the lower the variability of the model prediction) and by the prediction instability, calibration instability, and individual MAPE instability plots (S1–S3 Figs).

**Table 3. Model composition and comparison. Multiple logistic regression in the development cohort. Model 1 includes only the hospital variable. Model 2 including hospital variables and anesthetic and surgical variables. In model 3, those without statistical significance were excluded. In the final model, in addition to all variables with statistical significance, we included the interaction between airway surgery and tracheal intubation.**

| Model Composition | | |
|---|---|---|
| **Variables** | **OR (CI95%)** | **p value** |
| **Model 1 (n = 1339)** | | |
| Hospital | 1.75 (1.25–2.44) | <0.001 |
| **Model 2 (n = 1190)** | | |
| Hospital | 1.42 (0.93–2.18) | 0.102 |
| Age < 1 y | 1.91 (1.22–3.00) | 0.005 |
| ASA-PS 1 | Ref. | |
| ASA-PS 2 | 1.23 (0.78–1.94) | 0.374 |
| ASA-PS 3 | 1.84 (1.11–3.06) | 0.018 |
| ASA-PS 4 | 1.44 (0.35–6.01) | 0.617 |
| Obesity | 0.69 (0.28–1.72) | 0.427 |
| Lung or airway disease | 1.82 (1.20–2.76) | 0.005 |
| Current URI | 2.63 (1.32–5.23) | 0.006 |
| URI < 6 weeks | 1.55 (0.94–2.56) | 0.084 |
| Passive smoking | 1.07 (0.74–1.55) | 0.712 |
| History of prematurity | 2.28 (1.54–3.38) | <0.001 |
| Airway surgery | 1.71 (1.06–2.75) | 0.028 |
| Urgency | 1.54 (0.98–2.41) | 0.063 |
| Endotracheal intubation | 1.56 (1.05–2.31) | 0.027 |
| Premedication | 1.00 (0.67–1.51) | 0.988 |
| Laryngotracheal topical anesthesia | 0.89 (0.51–1.55) | 0.691 |
| Endovenous induction | 0.83 (0.56–1.23) | 0.354 |
| **Model 3 (n = 1339)** | | |
| Age < 1 y | 2.32 (1.59–3.39) | <0.001 |
| Lung or airway disease | 2.28 (1.63–3.18) | <0.001 |
| Current URI | 3.71 (2.06–6.66) | <0.001 |
| History of prematurity | 2.56 (1.80–3.64) | <0.001 |
| Airway surgery | 1.38 (0.97–1.97) | 0.069 |
| Endotracheal intubation | 1.67 (1.17–2.39) | 0.004 |
| **Final Model (n = 1339)** | | |
| Age < 1 y | 2.08 (1.41–3.05) | <0.001 |
| Lung or airway disease | 2.17 (1.55–3.18) | <0.001 |
| Current URI | 3.70 (2.06–6.66) | <0.001 |
| History of prematurity | 2.56 (1.80–3.63) | <0.001 |
| Endotracheal intubation*airway surgery | | 0.016 |
| General surgery | | |
| Without endotracheal intubation | Ref. | <0.001 |
| With endotracheal intubation | 2.13 (1.71–2.55) | |
| Airway surgery | | |
| Endotracheal intubation = 0 | Ref. | 0.459 |
| Endotracheal intubation = 1 | 0.76 (0.04–1.48) | |
| Without endotracheal intubation = 0 | | |
| Airway surgery = 0 | Ref. | 0.003 |
| Airway surgery = 1 | 3.15 (2.41–3.91) | |

*(Continued)*

**Table 3.** (Continued)

| Model Composition | | |
| --- | --- | --- |
| **Variables** | **OR (CI95%)** | **_p_ value** |
| Endotracheal intubation = 1 | | |
| Airway surgery = 0 | Ref. | 0.555 |
| Airway surgery = 1 | 1.13 (0.73–1.52) | |

ASA-PS, American Society of Anesthesiologists physical status; OR, odds ratio; CI, confidence interval; URI, upper respiratory infection.

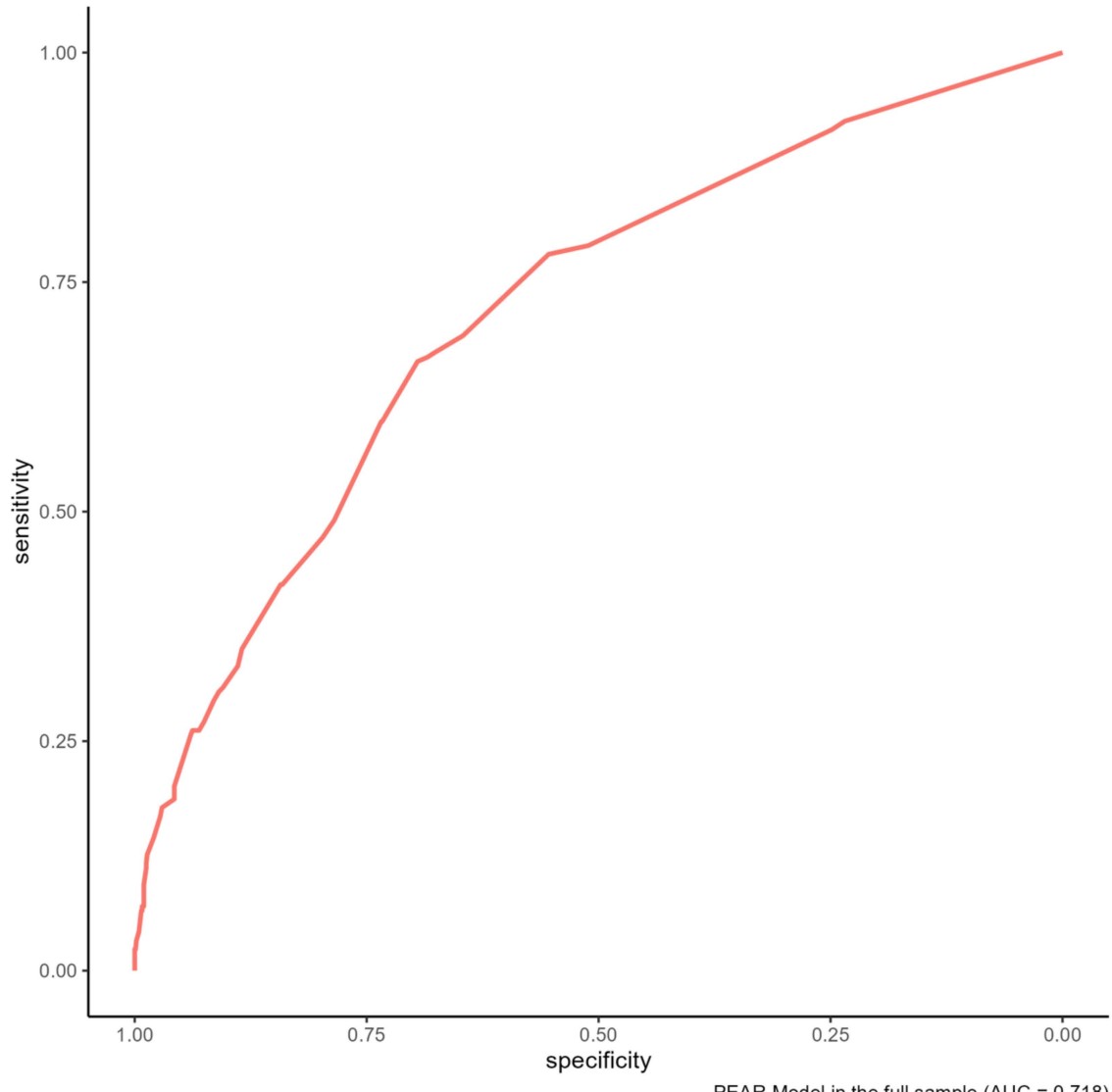

PEAR Model in the full sample (AUC = 0.718)

**Fig 1. Receiver operating characteristic curves of the PEAR model, demonstrating AUROC 0,71.**

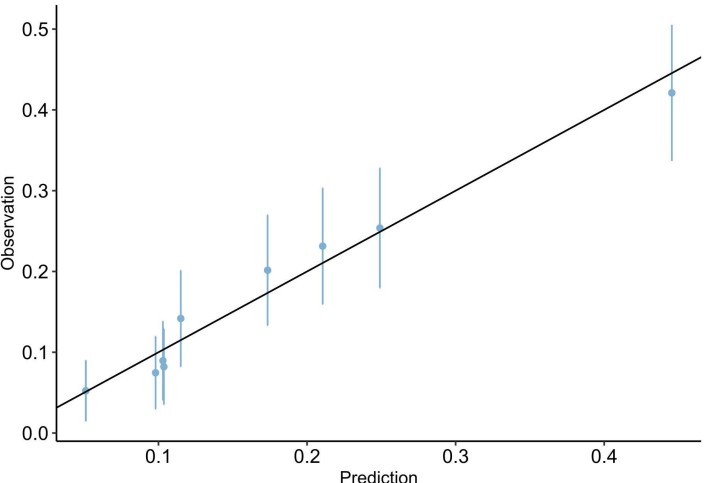

**Fig 2. Calibration plot of observed vs. predicted perioperative respiratory adverse events for derivation cohort.** Markers represent the observed perioperative adverse event rate (position on the Y-axis) in relation to the predicted perioperative adverse event rate (position on the X-axis); error bars, 95% confidence interval for the observed complication rate; solid lines, perfect calibration (i.e., observed=predicted). The circles correspond to the proportion of patients at each risk level.

## Comparison with existing risk scores

Fig 3 illustrates the ROC curve of the PEAR risk model (AUROC 0.69) and compares it with the COLDS score (AUROC 0.62) applied to all patients younger than six years of age in our population. The significant difference indicated the favorability of our model according to the DeLong test (p=0.004).

## Clinical usefulness

Considering that the prevalence of PRAEs in the general population is approximately 15% [11], we categorized the probability of PRAEs into three easily applied bedside classes: low risk, <15%; moderate risk, 15–30%; and high risk, >30%.

Once we stratified the risk for PRAEs - low, moderate, and high-risk – we identified that 37.2% of the patients had a higher risk for respiratory complications than the general population [11]. (S4 Fig).

We developed an interface in the R program for calculating the probability of complications, which can be used for research purposes as well as to support, facilitate, and share decisions before surgery. The calculator is available at https://isabelasirtoli.shinyapps.io/app_prae.

## Discussion

We prospectively evaluated the risk factors for perioperative respiratory adverse events in children undergoing surgery in two public Brazilian hospitals. Perioperative respiratory adverse events occurred in almost 16% of our sample and led to some negative effects in one out of every ten patients who experienced complications. We developed and validated a model for predicting the risk for perioperative respiratory adverse events using a few variables related to the patient, anesthesia, and surgical procedure. Our model presented better performance than previously reported models built for children.

Most complications in the pediatric surgical population involve the respiratory system [3], with an incidence varying from 2.8% to 26.2% [3,11,13,16,18] We reported a higher incidence of PRAE compared to large high-income cohorts [3,18], similar levels than Australian cohort [11], but lower than Ethiopian cohort [13], with similar designs. Particularities of the Brazilian public health system could explain these numbers. The fragmentation of care between primary and tertiary assistance and an overcrowded pediatric surgical agenda, along with the unmet need to increase both the volume and capacity of services,

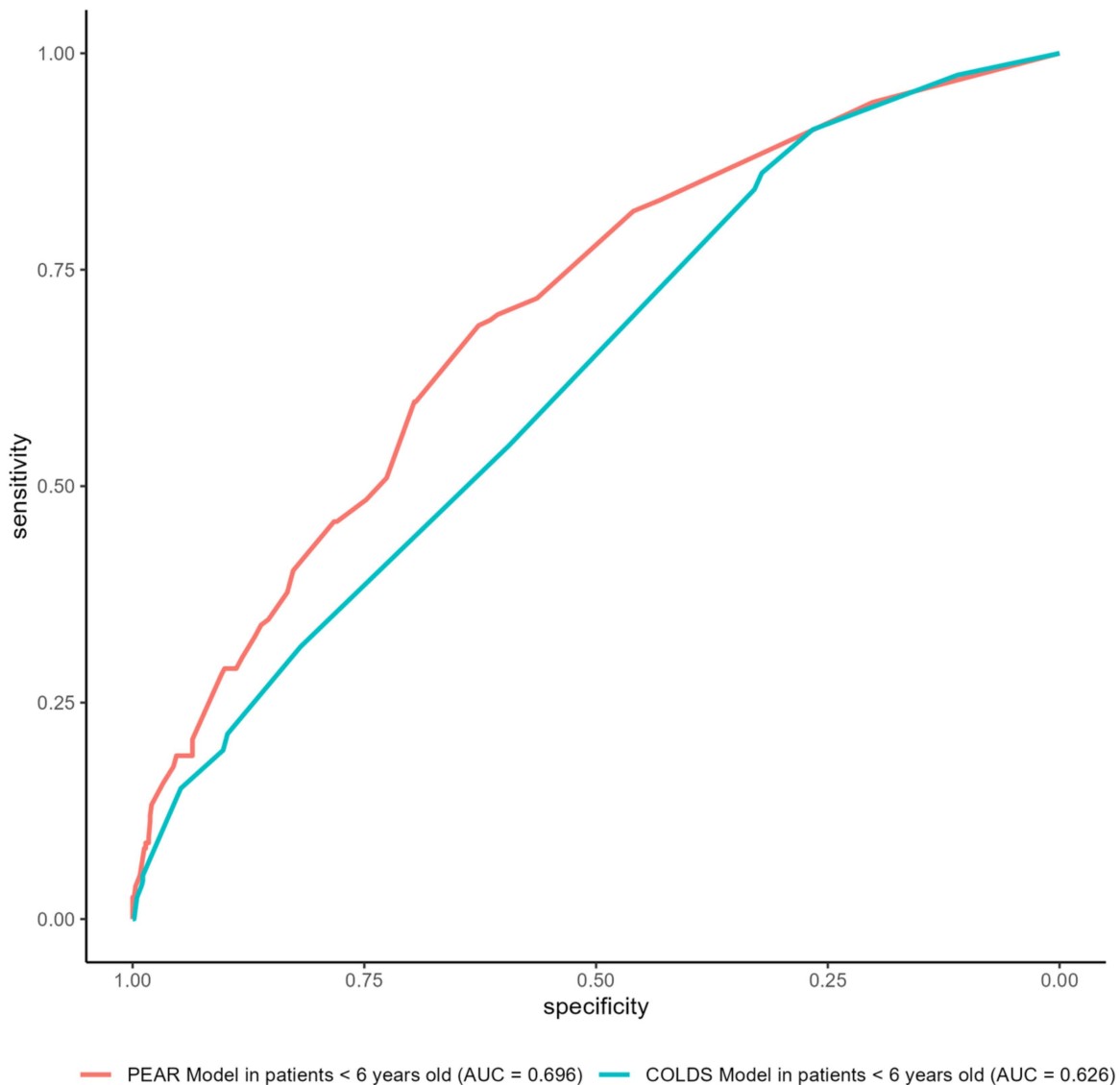

**Fig 3. Receiver operating characteristic curves of the PEAR model and the COLDS score, respectively demonstrating AUROC 0,71 and 0,63.**

might contribute to operating on children in suboptimal health conditions. However, caution is needed when comparing data-sets, as they can be influenced by outcome definitions, the study design, the nature of the surgeries, and the professionals' experience. Additionally, unmeasured factors such as health assessment, effects of environmental pollutants [31], hospital structure, care processes [32,33], and socio-economic factors [34] are increasingly recognized as influential on outcomes.

Our most important contribution was that we derived and validated a multivariable model highlighting variables associated with complications that could ground more comprehensive and interventions studies to help organizational and multiprofessional bedside decisions regarding risk of postoperative complications.

The final model for perioperative respiratory adverse events included the following variables: age younger than one year, lung or airway disease, URI at the time of surgery, history of prematurity, and interaction between airway surgeries and tracheal intubation. URI at the time of surgery was the most significant predictor, despite a low prevalence of this comorbidity in our sample (4.6% of prevalence vs. 25% [11] in the literature). Following a recent upper respiratory

infection, airway inflammation, interaction with the autonomic system, and consequent airway sensitization, which can persist for several weeks, account for the elevated risk in this population [11,35–37]. A possible explanation of the low prevalence of upper respiratory infection was that we included patients who underwent surgery during the COVID-19 pandemic, a period during which we reduced the number of elective surgeries and seldom recommended them for symptomatic patients. The presence of pulmonary or airway diseases was an independent risk factor for adverse perioperative respiratory events. Other studies also confirmed the relation of increased risk of adverse perioperative respiratory events with hypersensitivity of the airways caused by chronic or inflammatory diseases [3,9,11,38]. Likewise, due to specific anatomical and physiological conditions, younger children tend to develop adverse perioperative respiratory events [19]. Children <37 weeks of gestation are at risk for pulmonary and neurological sequelae, and the exact age at which this risk resolves is unknown [39]. In a robust study, Habre et al. showed that young age predicts severe respiratory complications and cardiovascular events. This fact was corroborated in our cohort where prematurity was strongly associated with adverse perioperative respiratory events [3].

The influence of the airway device in PRAE was previously studied [40], and the use of supraglottic device when compared to tracheal intubation was associated with a reduced incidence of PRAE [12]. Our study provide significant additional information regarding the impact of the invasiveness of the airway device, the type of surgery, and the interaction between these variables on the risk of adverse perioperative respiratory events [11,24,37,41]. We found an increased risk of adverse perioperative respiratory events with tracheal intubation, except in surgeries involving the airways. When the procedure involved the airway, the absence of intubation was associated with a higher risk of adverse perioperative respiratory events. A plausible explanation is the protection that intubation provides against surgical manipulation of airways, which can lead to oedema, bleeding, and mobilization of secretions, especially in the presence of hyperreactivity [42]. This finding is particularly important as it can support shared decision-making about airway devices in such challenging procedures for anesthesiologists and surgeons.

Based on our findings, several anesthetic planning strategies could help reduce PRAE in children identified as being at high risk. For these patients, premedication with inhaled salbutamol and the involvement of an experienced pediatric anesthesia team should be considered. Depending on the child's profile, it may be beneficial to avoid sedative premedication or opt for α2-agonists. Intravenous propofol remains the preferred choice for induction, with maintenance via TIVA. The use of desflurane should be avoided, and whenever possible, less invasive airway techniques, such as supraglottic airway devices, face masks, or high-flow nasal oxygen, may further mitigate the risk. In patients with severe bronchial hyperreactivity, especially when symptoms are pronounced and surgery cannot be delayed, perioperative steroids may be necessary, ideally administered at least 4–6 hours before the procedure [43].

The PEAR risk model performed well, with an AUROC of 0.71, and it is the first model to identify the risk of PRAE in Latin America. In the international context, the many existing studies [3,11,16,44] have not yet met the needs for practical use as instruments to help the decision-making process. Our model successfully identified risk factors for PRAE in Brazilian children with a good predictive performance compared to other scores validated in high-income countries [18,19]. Subramanyam et al. described a risk score for PRAEs in outpatient surgeries that performed well (AUROC = 0.71) [18]. However, their retrospective cohort study found a low incidence of respiratory events as it only considered outpatient surgeries and did not follow up with patients in the PACU [18]. Additionally, they did not include short-term modifiable risk factors, such as URI or the anesthetic technique used [18]. Compared to the COLDS score, the most internationally used tool, we demonstrated that our model's performance was superior. Moreover, the PEAR risk model can be applied to children up to 16 years old, includes simplified and combined variables related to the patient, surgery, and anesthesia and has lower inter-examiner variability since all variables are binary and easily defined.

## Strengths and limitations

We identified several advantages of our model. It is derived from a large prospective cohort of children who underwent surgery in two major public hospitals in Brazil. These children received various anesthetic techniques and underwent

surgeries of varying complexity and perioperative care, enhancing the representativeness of our population. Our model incorporates simple and intuitive variables related to at-risk children, making it easier for professionals to implement at bedside. We used contemporary statistical assessments to develop and validate the model, including robust performance, internal validation, and clinical stability measurements and also the comparison with other risk models. Finally, we proposed an open digital interface to calculate the risk categories to add information to the decision-making processes in childcare and to facilitate communication between professionals and families.

However, our model has some limitations. Firstly, although its performance was adequate, it has not been validated in other settings, and we emphasize that future work will focus on validating the model prospectively in different hospitals and children's populations, which will be essential to strengthen general acceptance and applicability across diverse clinical settings

Secondly, while the focus on immediate postoperative complications might not capture long-term consequences, this limitation is partly justified by our pragmatic study design, which facilitated data collection from many patients using a simple form in a clinical setting.

An additional limitation of our study is that data collection coincided with the COVID-19 pandemic. During this period, perioperative workflows, infection-control protocols, and institutional pathways for managing respiratory symptoms were substantially modified. Children presenting for anesthesia may also have had altered respiratory profiles due to pandemic-related factors, including reduced viral circulation, changes in exposure patterns, or residual effects following COVID-19 infection. These elements could have influenced both the baseline risk of PRAE and the perioperative decision-making processes. As a result, caution is warranted when extrapolating our findings to non-pandemic conditions. Further validation in post-pandemic cohorts will be necessary to confirm the stability and generalizability of our predictive model.

## Conclusion

There is a need to strengthen surgical pediatric care, especially in settings with limited resources [45]. In our Brazilian cohort, one in six pediatric patients who underwent surgery experienced a postoperative adverse pulmonary event, and one in ten of those who experienced PRAE faced some negative complications. Our study primarily identified the risk factors for pulmonary complications and developed a model to help identify vulnerable patients and plan the pediatric surgical journey more safely. However, the threshold of risk considered unsafe remains unclear. Further high-level studies are needed to externally validate the risk model and determine if interventions, such as the level of care or the type of airway device used, can reduce postoperative complications.

## Supporting information

**S1 Table. Detailed variables and outcomes description.**
(DOCX)

**S2 Table. Observed vs expected perioperative respiratory adverse events for deciles of risk.**
(DOCX)

**S3 Table. Logistic Regression Model for PRAE considering the hospital as a random effect.** Model 1 includes only the hospital variable. Model 2 including hospital variables and anesthetic and surgical variables. In model 3, those without statistical significance were excluded.
(DOCX)

**S1 Fig. The graph shows the prediction of instability for the original model and the 1000 bootstrap samples.** In it we observe the dispersion of the predicted values $B$ (y-axis) for each individual in relation to their original predicted value (x-axis).
(DOCX)

**S2 Fig. The calibration instability plot examines the instability in the calibration curves for the bootstrap models when evaluated on the original dataset.** In it, the *B* curves are overlaid on the same plot, along with the original calibration curve of the original model applied to the original data.
(DOCX)

**S3 Fig. APE can be shown graphically in a MAPE instability plot, which is a scatter of the MAPE value (y-axis) for each individual against their estimated risk from the original prediction model (x-axis).** This plot reveals the range of MAPE values and helps to identify if and where instability is of most concern for the original predictions.
(DOCX)

**S4 Fig. Incidence of perioperative respiratory adverse events according to each risk class.**
(DOCX)

**S5 Fig. Study diagram.**
(DOCX)

## Acknowledgments

We are greatful to hospital staff across Hospital de Clínicas de Porto Alegre e Hospital Conceição. Moreover, the authors thank the Programa de Pós-Graduação em Medicina: Ciências Médicas da Universidade Federal do Rio Grande do Sul.

**Presentation**

Preliminary data from the study have been presented at several scientific meetings. Among them: oral presentation at the ASA Meeting 2022 (session OR15−3 – Pediatric Airway/Respiratory), 12th European Congress for Pediatric Anesthesiology in 2022 (panel 6 – Best abstracts session, ID 34) and 13th European Congress for Pediatric Anesthesiology in 2023 (panel 6: Best abstracts, ID 9), Euroanesthesia 2023 (presentation number 05AP04−02, abstract reference AS-ESAIC-2023–00213).

## Author contributions

**Conceptualization:** Isabela S. Sirtoli, Luciana C. Stefani.

**Data curation:** Isabela S. Sirtoli, Stela J. Castro, Luciana C. Stefani.

**Formal analysis:** Isabela S. Sirtoli, Stela J. Castro, Luciana C. Stefani.

**Funding acquisition:** Luciana C. Stefani.

**Investigation:** Isabela S. Sirtoli, Cleiton Pando, Clarissa Mendanha.

**Methodology:** Isabela S. Sirtoli, Paulo C. S. Neto, Cleiton Pando, Clarissa Mendanha, Luciana C. Stefani.

**Project administration:** Isabela S. Sirtoli, Cleiton Pando, Luciana C. Stefani.

**Resources:** Isabela S. Sirtoli, Paulo C. S. Neto, Luciana C. Stefani.

**Software:** Raissa M. Scorsatto.

**Supervision:** Isabela S. Sirtoli, Stela J. Castro.

**Validation:** Isabela S. Sirtoli, Stela J. Castro.

**Visualization:** Isabela S. Sirtoli.

**Writing – original draft:** Isabela S. Sirtoli, Paulo C. S. Neto, Luciana C. Stefani.

**Writing – review & editing:** Isabela S. Sirtoli, Stela J. Castro, Luciana C. Stefani.

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
