## [Decision Letter · Decision Letter 0]

10 Nov 2025

Dear Dr. Stefani,

Thank you for submitting your manuscript to PLOS ONE. After careful consideration, we feel that it has merit but does not fully meet PLOS ONE’s publication criteria as it currently stands. Therefore, we invite you to submit a revised version of the manuscript that addresses the points raised during the review process.

We look forward to receiving your revised manuscript.

Kind regards,

Sathyaprasad Burjonrappa

Academic Editor

PLOS ONE

Journal Requirements:

https://journals.plos.org/plosone/s/file?id=ba62/PLOSOne_formatting_sample_title_authors_affiliations.pdf....

**2.** Thank you for stating in your Funding Statement:Thank you for stating in your Funding Statement:Thank you for stating in your Funding Statement:Thank you for stating in your Funding Statement:

“This work was supported by Financiamento de Incentivo à Pesquisa do Hospital de Clínicas de Porto Alegre [FIPE-HCPA Project 2020-0363] and and the Foundation for Research of the State of Rio Grande do Sul – FAPERGS - PPSUS 2020 [Secretaria de Saúde do Estado do Rio Grande do Sul – SES-RS, Ministério da Saúde – MS – Departamento de Ciência e Tecnologia da Secretaria de Ciência, Tecnologia, Inovação e Insumos Estratégicos em Saúde – Decit/SCTIE/MS and Conselho Nacional de Desenvolvimento Científico e Tecnológico – CNPq]”

“This work was supported by Financiamento de Incentivo à Pesquisa do Hospital de Clínicas de Porto Alegre [FIPE-HCPA Project 2020-0363] and and the Foundation for Research of the State of Rio Grande do Sul – FAPERGS - PPSUS 2020 [Secretaria de Saúde do Estado do Rio Grande do Sul – SES-RS, Ministério da Saúde – MS – Departamento de Ciência e Tecnologia da Secretaria de Ciência, Tecnologia, Inovação e Insumos Estratégicos em Saúde – Decit/SCTIE/MS and Conselho Nacional de Desenvolvimento Científico e Tecnológico – CNPq]”

4. In the online submission form, you indicated that [Insert text from online submission form here].

6. Please be informed that funding information should not appear in the Acknowledgments section or other areas of your manuscript. We will only publish funding information present in the Funding Statement section of the online submission form. Please remove any funding-related text from the manuscript.

Additional Editor Comments (if provided):

The authors have done an excellent job of coming up with factors that increase risk for PRAE in their setting. I would like to see a paragraph added to the discussion about how they would use their results to change their preoperative process for infants and children undergoing anesthesia. We all know that surgical procedures in the pediatric population carry risk but mitigating that risk is not easy particularly if it is an emergent or urgent procedure. This would be particularly helpful to the readers in settings similar to where the authors practice: Specifically, I would ask if delayed extubation, perioperative airway medications or perhaps steroids, avoiding drugs that can irritate the airways are considerations?

Secondly while internal validation (using the bootstrap model) is better than no validation why did the authors not try an external validation method? It would allow for more general acceptance of the data than to their own/similar cohort.

Reviewers' comments:

Reviewer's Responses to Questions

**Comments to the Author**

1. Is the manuscript technically sound, and do the data support the conclusions?

Reviewer #1: Yes

Reviewer #2: Yes

2. Has the statistical analysis been performed appropriately and rigorously?

Reviewer #1: I Don't Know

Reviewer #2: Yes

3. Have the authors made all data underlying the findings in their manuscript fully available?

Reviewer #1: Yes

Reviewer #2: Yes

4. Is the manuscript presented in an intelligible fashion and written in standard English?

Reviewer #1: Yes

Reviewer #2: Yes

Reviewer #1: Thank you for giving me the privilege to review your manuscript. I thoroughly enjoyed reading your paper and am excited to be part of this process. Your study results, data are compelling and support the clinical acumen we use in decisions surrounding preoperative surgical clearance in pediatrics. It is exciting to have data that would support the decision making. My comment on statistical analysis does not mean that I do not agree with your data but strictly is my omission of data that I do not have expertise on.

Reviewer #2: The manuscript is scientifically sound. It is easy to read and helps address important side effects of peri-operative time frame in pediatric patients. There can be consideration for controlling for pre-existing illnesses to improve the generalizability of this predictive model. The time frame of this study being conducted during the covid-19 pandemic is also a limitation to peri operative anesthetic adverse events.

.

Reviewer #1: No

Reviewer #2: No

---

## [Author Response · Author response to Decision Letter 1]

9 Mar 2026

Review Comments to the Author

Editor Considerations

Consideration 1: The authors have done an excellent job of coming up with factors that increase risk for PRAE in their setting. I would like to see a paragraph added to the discussion about how they would use their results to change their preoperative process for infants and children undergoing anesthesia. We all know that surgical procedures in the pediatric population carry risk but mitigating that risk is not easy particularly if it is an emergent or urgent procedure. This would be particularly helpful to the readers in settings similar to where the authors practice: Specifically, I would ask if delayed extubation, perioperative airway medications or perhaps steroids, avoiding drugs that can irritate the airways are considerations?

Author: We thank the editor for his comments. We agree that discussing how our findings may influence and improve our preoperative processes for children undergoing anesthesia would add meaningful value for readers, particularly those practicing in similar contexts. In response to this suggestion, we have added a paragraph to the Discussion section - page 21:

"Based on our findings, several anesthetic planning strategies could help reduce PRAE in children identified as being at high risk. For these patients, premedication with inhaled salbutamol and the involvement of an experienced pediatric anesthesia team should be considered. Depending on the child's profile, it may be beneficial to avoid sedative premedication or opt for α2-agonists. Intravenous propofol remains the preferred choice for induction, with maintenance via TIVA. The use of desflurane should be avoided, and whenever possible, less invasive airway techniques, such as supraglottic airway devices, face masks, or high-flow nasal oxygen, may further mitigate the risk. In patients with severe bronchial hyperreactivity, especially when symptoms are pronounced and surgery cannot be delayed, perioperative steroids may be necessary, ideally administered at least 4–6 hours before the procedure."

Consideration 2: Secondly while internal validation (using the bootstrap model) is better than no validation why did the authors not try an external validation method? It would allow for more general acceptance of the data than to their own/similar cohort.

Author: We thank the reviewer for this observation regarding model validation. We agree that external validation is highly valuable and can enhance the generalizability of predictive models beyond the original study cohort. We used the bootstrapping method for internal validation because it is a more modern statistical method, avoiding dividing our sample into separate sets and generating unstable models. Bootstrapping uses the entire sample repeatedly, preserving statistical power. Bootstrapping better estimates the model's optimism (overfitting), and is considered the gold standard for internal validation of predictive models. To address this point, we have expanded the Discussion section to clarify why external validation could not be performed - page 22-23:

"We emphasize that future work will focus on validating the model prospectively in different hospitals and children’s populations, which will be essential to strengthen general acceptance and applicability across diverse clinical settings”.

Reviewer 1 considerations

Consideration 1: Thank you for giving me the privilege to review your manuscript. I thoroughly enjoyed reading your paper and am excited to be part of this process. Your study results, data are compelling and support the clinical acumen we use in decisions surrounding preoperative surgical clearance in pediatrics. It is exciting to have data that would support the decision making. My comment on statistical analysis does not mean that I do not agree with your data but strictly is my omission of data that I do not have expertise on.

Author: We thank the reviewer for the encouraging feedback and for the thoughtful evaluation of our work. We appreciate your comments regarding the statistical analysis and fully understand the challenges that this aspect may present. To facilitate greater clarity and transparency, we would be pleased to make our anonymized data available, should this assist in further understanding the statistical methods or in reproducing our analyses. We hope this contributes to strengthening confidence in our findings and supports continued dialogue around the study’s methodology.

Reviewer 2 consideration

Consideration 1: The manuscript is scientifically sound. It is easy to read and helps address important side effects of peri-operative time frame in pediatric patients. There can be consideration for controlling for pre-existing illnesses to improve the generalizability of this predictive model.

Author: We thank the reviewer for their positive assessment of our manuscript. We appreciate the suggestions provided. Regarding the recommendation to control for pre-existing illnesses to enhance the generalizability of the predictive model, we agree that this is an important consideration. In the initial stages of model construction, we evaluated the inclusion of ASA physical status classification as a global marker of baseline health. However, because our goal was to develop a practical and parsimonious model suitable for real-world clinical use, ASA was ultimately excluded during model refinement. Importantly, our final model incorporates specific pre-existing conditions—such as pulmonary comorbidities and a history of prematurity—which directly capture clinically relevant chronic health factors in pediatric patients. Nevertheless, we recognize that our model is parsimonious and, because of this, it might not include all factors related to pulmonary complications, such as broader baseline health conditions and social determinants of health. Future studies incorporating real-world data and leveraging large language models may help address these questions and further refine predictive accuracy.

Consideration 2: The time frame of this study being conducted during the covid-19 pandemic is also a limitation to perioperative anesthetic adverse events.

Author: We also acknowledge the reviewer’s point about the study period overlapping with the COVID-19 pandemic. This is indeed a relevant limitation, as peri-operative practices and patient profiles may have been influenced during this time. We have now explicitly addressed this issue in the Discussion - page 23.

"An additional limitation of our study is that data collection coincided with the COVID-19 pandemic. During this period, perioperative workflows, infection-control protocols, and institutional pathways for managing respiratory symptoms were substantially modified. Children presenting for anesthesia may also have had altered respiratory profiles due to pandemic-related factors, including reduced viral circulation, changes in exposure patterns, or residual effects following COVID-19 infection. These elements could have influenced both the baseline risk of PRAE and the perioperative decision-making processes. As a result, caution is warranted when extrapolating our findings to non-pandemic conditions. Further validation in post-pandemic cohorts will be necessary to confirm the stability and generalizability of our predictive model."

---

## [Editor Report · Decision Letter 1]

1 Apr 2026

Incidence and Risk Factors of Perioperative Respiratory Adverse Events in Pediatric Surgical Patients: Development and Validation of a Predictive Model in Brazil

PONE-D-25-40901R1

Dear Dr. Stefani,

We’re pleased to inform you that your manuscript has been judged scientifically suitable for publication and will be formally accepted for publication once it meets all outstanding technical requirements.

Kind regards,

Sathyaprasad Burjonrappa

Academic Editor

PLOS One

Additional Editor Comments (optional):

Accept
---

## [Editor Report · Acceptance letter]

PONE-D-25-40901R1

PLOS One

Dear Dr. Stefani,

I'm pleased to inform you that your manuscript has been deemed suitable for publication in PLOS One. Congratulations! Your manuscript is now being handed over to our production team.

Kind regards,

on behalf of

Dr. Sathyaprasad Burjonrappa

Academic Editor

PLOS One